# A DUAL-METRIC APPROACH FOR MODEL SELECTION IN SELF-SUPERVISED LEARNING FOR HISTOPATHOLOGY

## ABSTRACT

Selecting appropriate models during self-supervised training of vision transformers in histopathology is challenging. Recent efforts to quantify the quality of self-supervised learning representations through rank estimation approaches have shown promise in natural image classification tasks. However, their effectiveness in histopathology, particularly for non-linear tasks such as instance segmentation and classification from whole slide images, remains unexplored. This study proposes an approach for model selection in histopathology by combining task-specific metrics (such as accuracy) and task agnostic metrics (such as rank estimation). This work shows that by training several small-scale histopathology models and applying the proposed model selection approach, one can achieve instance segmentation performance comparable to state-of-the-art models trained on much larger datasets. The proposed approach also allows for obtaining a model based on the type of downstream task. Towards this end, three types of model selection based on the downstream task performance were evaluated: classification-best, segmentation-best, and a best all-round one. When evaluated on held-out classification and weakly supervised learning tasks, the most performant checkpoints often occur earlier in training, indicating potential performance saturation mid way in the training for histopathology models. These results highlight the importance of appropriate model selection for self-supervised learning in histopathology.

## 1 INTRODUCTION

### 1.1 MOTIVATIONS

Mirroring advancements in self-supervised pre-training of vision transformers (ViT) (Dosovitskiy et al., 2020) for natural images, a significant number of models have been proposed in computational pathology literature. As transformers significantly benefit from scaling in dataset size and compute (Kolesnikov et al., 2020), histopathology models are increasingly getting larger, with proportional increases in the amount of data, compute and therefore the associative power consumption required for training these models.

Despite considerable interest and progress of self-supervised learning (SSL) in histopathology, there is an important and often overlooked challenge: estimating the the generalizability of a model in order to terminate training is challenging. This is because minimizing the typical training objective in self-supervised learning may not translate to better downstream task performance (Geiping et al., 2023). Furthermore, as label-free learning algorithms strive for generality in a diverse landscape of downstream tasks, finalizing on a predefined set of training iterations may trade balance in favor of one type of downstream task over another, such as trading image classification performance over image segmentation performance. Under current practices in histopathology literature, gauging model efficacy is done based on tile-level (patches[1] extracted from scanned tissue images, also called whole slide images, or WSI) benchmark tasks, and slide-level benchmark tasks. But does benchmark performance correlate with downstream task performance? Is one kind of benchmark

---

[1] patches here are typically 224×224 pixel non-overlapping sub-sections of a larger image, and is distinct from patches used in vision transformer, which typically is around 16×16.

task suitable over the other? As of writing this article, answers to these questions remain unclear. Some effort to quantify the quality of SSL representations, particularly by estimating the rank of representations, have been made in natural image literature (briefly elaborated in §2.2). The main idea here is that there are some salient, desirable properties that representations must have in order to be beneficial to downstream tasks, thus making the assumption that these quality metrics directly correlate with downstream performance. Yet, they have only thus far been tested on linear probing tasks, which indeed benefit from an appropriate structure within the representation's eigenvectors, such as its eigenspectrum decay for instance, or its entropy. Therefore, their usefulness on other tasks, particularly instance segmentation tasks that are crucial to histopathology, is not known.

In the absence of theoretical frameworks that explore stopping criteria in self-supervised learning approaches, the core thesis of this work is that the appropriate approach is a combination of out-of-distribution benchmark performance, which is termed task-dependent metrics, and representation quality estimation approaches, which is termed task-agnostic metrics [2]. Therefore, this work attempts to build a bridge between these two topics by proposing a simple but effective model selection procedure that give improved performance on downstream tasks.

## 1.2 CONTRIBUTIONS

This study proposes a model selection procedure for field of histopathology by combining publicly available histopathology benchmark task-specific metrics and representation quality based task-agnostic metrics to propose a simple approach for model selection for self-supervised histopathology models. The approach is described in §3.

Several encoders are trained by varying the dataset, model size, and model architecture, in order to identify appropriate checkpoints for downstream clinical use using the proposed approach. The model selection is segregated into three approaches based on the type of tile-level tasks: a classification-best checkpoint obtained using only classification benchmarks, a segmentation-best checkpoint on benchmark tasks based only on instance segmentation benchmarks, and a task-agnostic model that provides favorable performance on both types of tasks. The performance of each checkpoint type is investigated on out-of-distribution benchmark tasks in §5.2 and slide-level tasks in §5.3.

The code and accompanying task-specific and task-agnostic metric data for all of our experiments will be released with this paper.

## 1.3 SCOPE

The encoders trained in this work are kept small-scale, since training data from multiple tissue-types can become prohibitive in the training stage. Therefore, only one tissue type is chosen, i.e., those obtained from patients with Lung Adenocarcinoma (LUAD), a variant of Non-Small Cell Lung Cancers (NSCLC) (Kundra et al., 2021). The classification of the Epithelial Growth Factor Receptor (EGFR) biomarker serves as a held-out downstream task at the slide-level, alongside the prediction of LUAD subtypes at the patch level, while several publicly available benchmarks (introduced in appendix A) are utilized as surrogate tasks to study model convergence. As these public datasets often contain data from diverse cancer types, studying benchmarking performance explicitly measures out-of-distribution generalization of the relational-distribution type (Farquhar & Gal, 2022), which is useful in determining the efficacy of the proposed model selection approach. The scope of this work is further limited by choosing the dinov1 self-supervised learning method (Caron et al., 2021) to train the vision encoders as this is a relatively simpler SSL framework compared to ones employed in training larger models with bigger batch sizes, for which approaches like dinov2 (Oquab et al., 2023) were designed.

---

[2]Here we use the relational distribution definition from Farquhar & Gal (2022) to define the out-of-distribution type of the benchmarks.

## 2 RELATED WORKS

### 2.1 SELF-SUPERVISED LEARNING IN DIGITAL HISTOPATHOLOGY

**Self-supervised learning approaches:** The surge in foundation model training for digital histopathology follows the successes of joint-embedding self-supervised learning (SSL) techniques in natural images (Geiping et al., 2023). The goal of these techniques is to embed and align two separately augmented variations of an image using various alignment objectives, such as the contrastive learning target used in Azizi et al. (2023); Ciga et al. (2022). This target utilizes the infoNCE loss (Oord et al., 2018) introduced in methods like SimCLR (Chen et al., 2020), where representations of image augmentations are aligned, or CLIP (Radford et al., 2021) and SigLIP (Zhai et al., 2023), where text and image pairs are aligned.

A common SSL framework used by many foundation models is the dinov1 (Caron et al., 2021) and dinov2 (Oquab et al., 2023) approach, both of which employ a teacher-student network. In this network, the teacher's representations are skewed from the student's using a centered softmax, and the teacher's weights are updated using an exponential moving average over the student. Dinov2 improves upon dinov1 in terms of training stability over larger batch sizes using the Ko-Leo regularizer (Sablayrolles et al., 2018) and by replacing the centering step in dinov1 with a Sinkhorn-Knopp centering step (Caron et al., 2020). Dinov2 also includes the iBOT loss (Zhou et al., 2021), which introduces patch-level losses in addition to the image-level losses present within Dinov1. The vanilla dino framework has been immensely successful in digital histopathology and forms the foundation for a majority of models trained on histopathology data (Zimmermann et al., 2024; Vorontsov et al., 2024; Chen et al., 2024; Saillard et al., 2024; Nechaev et al., 2024; Campanella et al., 2023; Dippel et al., 2024; Xu et al., 2024; Juyal et al., 2024).

Recently, modifications to the Dino framework have been proposed to cater to the specialized field of digital histopathology. For example, Zimmermann et al. (2024) replaced the Ko-Leo regularizer with the Kernel Density Estimator (KDE) and observed an increase in performance and stability, since a typical minibatch of digital histopathology images is relatively more similar compared to natural images spanning thousands of classes. Juyal et al. (2024) proposed merging masked image modelling with the vanilla Dino approach by including the Masked Auto-Encoder (MAE) loss objective (He et al., 2022). This is a similar style of including a patch-level modeling approach as in iBOT, but is an image modelling approach instead of a joint-embedding approach. Therefore, the loss is estimated over a reconstruction of the masked image with the input. In this work, they also introduce a Fourier reconstruction loss (Wang et al., 2024), which decomposes the Fourier transform of the reconstructed image from the MAE branch into low and high frequency components using a band-pass filter. This is perhaps included in order to counter a known issue with MAE which causes the representations to favor high-frequency components in an image over low-frequency ones (Bao et al., 2021; Ramesh et al., 2021). As these developments continue to progress, this work chose to use the original Dino framework by Caron et al. (2021), as modifications can successively be introduced to the learning approach later on. Training stability in the dinov1 approach can be achieved using small batch-sizes, which is ideal for the single-tissue scale of the experiments in this work.

**Multi-scale adaptation:** In a typical slide-evaluation procedure, histopathologists examine tissue samples at multiple scales. This is because at larger fields of view, the tissue architecture is distinct, while higher magnifications enables cellular features to be distinguished. Therefore, histopathology benchmarks tend to be distributed across magnifications, from 2 microns/px to 0.25 microns/px, the latter being more favorable for cell segmentation tasks. Histopathology models often exposed to increasingly larger resolutions of patches extracted from whole-slide images in the final stage of the pre-training (Chen et al., 2024). Recently that some works have considered the multi-scale aspect of histopathology during pre-training. For example, the HIPT model (Chen et al., 2022) considers a hierarchical set of feature extractors on a series of varying patch sizes of the vision transformer patch embedding module ($16\times16$, $256\times256$, and $4096\times4096$) in order to capture cellular, tile level, and region level information. Juyal et al. (2024) use the FlexiViT architecture (Beyer et al., 2023) to introduce a range of scales into the encoder training. Finally, GigaPath (Xu et al., 2024) utilizes the LongNet architecture (Ding et al., 2023) as a decoder to produce slide-level embeddings from a set of tile-level embeddings extracted using a standard vision transformer backbone. These multi-scale adaptations are largely architectural, but a simpler approach is to pre-train vanilla architectures on multi-scale data, thus introducing data variability and the ability of the encoder to adapt to the

myriad of scales of downstream tasks. Published works include Kang et al. (2023) and Zimmermann et al. (2024), which randomly mix the FOV of the images during pre-training without changes to the image resolution. Kang et al. (2023) uses tiles extracted from magnifications of 0.25 and 0.5 $\mu$m/px, while Zimmermann et al. (2024) use 0.25, 0.5, 1, and 2 $\mu$m/px in their dataset. One of the benefits observed by Kang et al. (2023) seems to be improved convergence during training. But the most important benefit is that this enables the encoder to use a benchmark's native resolution and magnification for assessment, and therefore is the approach followed by this work.

## 2.2 TASK-AGNOSTIC QUALITY METRICS

Joint-embedding self-supervised learning (SSL) approaches train encoders solely at the representation level, making it challenging to predict when the training process has reached a level suitable for downstream tasks. To address this, recent research has focused on rank-based representation quality metrics, which operate under the assumption that optimal metric values will lead to improved benchmark performance, and have been reported to correlate with downstream performance. Garrido et al. (2023) introduced RankMe, which computes the Shannon entropy of the eigenvalues of a set of representations as the effective rank of the embedding matrix, serving as a proxy for representational power. RankMe demonstrated a strong correlation with downstream linear probing performance across various SSL methods and architectures. Thilak et al. (2023) extended RankMe by applying linear discriminant analysis (Martinez & Kak, 2001), estimating a generalized covariance matrix using representations of different images and transformed variants of the same image, particularly those used in the SSL method. They then estimate the entropy of the eigenvalues of the generalized covariance matrix, capturing the representation behavior explicitly as determined by the SSL objective. Building on theoretical insights, Agrawal et al. (2022) proposed $\alpha$-ReQ, which measures the decay rate of the eigenspectrum of the representation covariance matrix, arguing that an optimal rate balances expressiveness and generalization. However, as noted by Thilak et al. (2023), $\alpha$-ReQ is sensitive to linear transformations that arbitrarily influence the eigenspectrum matrix, allowing for high rank registration despite potential degradation in downstream performance.

One of the key advantages of rank-based representation quality metrics is their ability to measure dimensional collapse, where one eigenvalue dominates while others contribute minimally towards the representation. This provides valuable insights into the expressiveness and generalization capabilities of the learned representations. However, a significant limitation of rank approximation as a quality metric is their reliance on the linear behavior of eigenvalues in representing the quality of learned features. While this linear approach may be suitable for linear-probing tasks, it may not be adequate for inherently non-linear tasks in the histopathology domain, such as multiple instance learning. Using these task-agnostic metrics in conjunction with task-specific metrics from a set of benchmark tasks can help mitigate the drawbacks of rank-estimation approaches, and is the track followed in this work.

## 3 MODEL SELECTION PROCEDURE

The approach described in Algorithm 1 is a simple process that identifies the best-performing checkpoint across a set of out-of-distribution benchmark tasks, yielding task-specific metrics such as the aggregated jaccard index (Kumar et al., 2019) or the weighted F1 metric, and task-agnostic representation quality metrics, such as RankMe, LiDAR, and $\alpha$-ReQ. Given $N$ task-specific metrics, $M$ task-agnostic metrics, and $E$ checkpoints saved during a particular run (e.g., every 5 epochs), an $N \cdot M$ number of samples are obtained, indicating the result between a performance metric for each task and its representation quality across the $E$ saved checkpoints.

The task-agnostic metrics (e.g., RankMe, LiDAR, or $\alpha$-ReQ, detailed in Appendix C) are calculated from the test set of the pre-training dataset, which is distinct from the benchmark performance dataset. Each task's result is determined by a normalized benchmark metric between 0 and 1, such as the aggregated Jaccard index for instance segmentation (Kumar et al., 2019) and the weighted F1 score for classification tasks[3]. To estimate the best checkpoint for each benchmark task-representation metric pair, the sum of the normalized representation metric (scaled to its range)

---

[3]In the classification tasks presented in this work, the output probability of the predicted classes is thresholded to maximize Youden's index Youden (1950), unless explicitly defined.

---

**Algorithm 1** Proposed Dual-Metric Model Selection Approach

---

**Inputs:** $U_1, \ldots, U_N$: Set of $N$ task-specific metrics,
$\quad$ $V_1, \ldots, V_M$: Set of $M$ task-agnostic metrics,
$\quad$ $\mathbf{P}^{ts} \in \mathbb{R}^{E \times N}$: Performance matrix of $E$ epochs for $N$ task-specific metrics,
$\quad$ $\mathbf{P}^{ta} \in \mathbb{R}^{E \times M}$: Performance matrix of $E$ epochs for $M$ task-agnostic metrics
**Output:** Selected epoch $e^*$
1: $\mathbf{N}^{ts} \in \mathbb{R}^{E \times N}$, where $N_{i,j}^{ts} = \underset{i}{\text{MinMaxNormalize}}(\mathbf{P}_{i,j}^{ts})$
$\quad \forall \ \ i \in [1, \ldots, E], j \in [1, \ldots, N]$ $\qquad\qquad\qquad$ ▷ Normalize task-specific metrics
2: $\mathbf{N}^{ta} \in \mathbb{R}^{E \times M}$, where $N_{i,j}^{ta} = \underset{i}{\text{MinMaxNormalize}}(\mathbf{P}_{i,j}^{ta})$
$\quad \forall \ \ i \in [1, \ldots, E], j \in [1, \ldots, M]$ $\qquad\qquad\qquad$ ▷ Normalize task-agnostic metrics
3: $\mathbf{C} \in \mathbb{R}^{N \times M}$, where $C_{i,j} = \underset{e}{\text{argmax}}(N_{e,i}^{ts} + N_{e,j}^{ta})$
$\quad \forall \ \ e \in [1, \ldots, E], i \in [1, \ldots, N], j \in [1, \ldots, M]$ $\quad$ ▷ Selected epoch for each metric pair
4: $S = \{\text{unique}(C_{i,j}) \mid i \in \{1, \ldots, N\}, j \in \{1, \ldots, M\}\}$ $\qquad$ ▷ Set of unique epochs from $\mathbf{C}$
5: $\mathbf{r} \in \mathbb{R}^{|S|}$, where $r_k = \sum_{j=1}^{N} N_{s_k, j}^{ts}$ for $s_k \in S$ $\quad$ ▷ Relative improvement summed over tasks
6: $e^* = C_{i^*, j^*}$, where $(i^*, j^*) = \underset{n}{\text{argmax}} \ r_n$ $\qquad$ ▷ Epoch with highest relative improvement
7: **return** $e^*$

---

and the benchmark value, with similar normalization, is maximized[4]. This process yields $N \cdot M$ models for each benchmark result-representation metric pair. To select a single checkpoint, the relative improvement for each of the $N \cdot M$ models is computed using the best benchmark value. The epoch with the highest average relative improvement across all tasks is chosen as the final model.

Three separate model selections are made: $e_a^*$, $e_c^*$, and $e_s^*$, representing the best checkpoint considering all benchmark tasks (both instance segmentation and classification benchmarks), the classification-best checkpoint, and the instance segmentation-best checkpoint, respectively. The performance of all three types of checkpoints on the downstream EGFR prediction task is studied, and remarks are made in the discussions.

## 4 EXPERIMENTS

The description of the pre-training, benchmarking, and downstream datasets can be found in Appendix A. The appendices also provide the pre-training, slide-level benchmarking, and downstream task training procedures in Appendix B. The steps taken in estimating task-agnostic representation quality metrics are described in Appendix C. To state briefly, the benchmark tasks considered here are of two types: classification (BACH Aresta et al. (2019), MHIST Wei et al. (2021), CRC Kather et al. (2018)), and nuclei instance segmentation (PanNuke Gamper et al. (2020), MoNuSeg Kumar et al. (2019)). The task-agnostic metrics used in this work are the LiDAR metric (Thilak et al., 2023), RankMe (Garrido et al., 2023), and $\alpha$-ReQ (Agrawal et al., 2022). The following briefly describes the models and their associated motivations.

Nine distinct models were trained using the vanilla dinov1 framework (Caron et al., 2021) for greater than 230 epochs, all utilizing Vision Transformer (Dosovitskiy et al., 2020) (ViT) backbones with four registers (Darcet et al., 2023). This included three ViT-B models and three ViT-S models, with variations in the number of magnifications in the data and, consequently, the number of images per epoch, as the reduction in the number of magnifications used in the dataset was not compensated by increasing the dataset size. Architectural variation was also introduced by implementing soft mixture-of-experts (Puigcerver et al., 2023), a model paradigm that allows increasing model capacity without sacrificing throughput. This was done at the ViT-S scale, varying the number of experts (4, 32, and 128) while maintaining one slot per expert, thus allowing variation in parameter count.

The models presented in this work, described in Table 1, ranged in size from 21.6M to 922.3M parameters, with training datasets varying from 3.27M to 10.25M images. Variation in the for-

---

[4]In the special case of $\alpha$-ReQ, the sum of the negative of the quality metric subtracted by 1 and the benchmark task performance is maximized, as the authors proposed that an optimal $\alpha$-ReQ value lies around this value.

Table 1: Tabulated details of a diverse set of vision transformer encoders trained using the Dino pretraining approach (Caron et al., 2021). *Magnification* column indicates dataset diversity and the *Encoder* columns indicate the base encoder. Soft mixture of experts models (Puigcerver et al., 2023) are indicated using the shorthand *SMoE* followed by a hyphen and an integer indicating the total number of experts in the feed forward layer of the transformer. †indicates models trained on single magnification dataset, including both 20× and 40× encoders.

| Model | Magnification | | | | Encoder | Params | Training |
|---|---|---|---|---|---|---|---|
| | 5× | 10× | 20× | 40× | | | Images |
| ViT-S† | | | ✓ | | ViT-S | 21.6M | 3.36M |
| ViT-S† | | | | ✓ | ViT-S | 21.6M | 3.27M |
| ViT-S | ✓ | | ✓ | ✓ | ViT-S | 21.6M | 10.25M |
| ViT-B† | | | ✓ | | ViT-B | 85.8M | 3.36M |
| ViT-B† | | | | ✓ | ViT-B | 85.8M | 3.27M |
| ViT-B | ✓ | | ✓ | ✓ | ViT-B | 85.8M | 10.25M |
| ViT-S SMoE-4 | ✓ | | ✓ | ✓ | ViT-S | 42.9M | 10.25M |
| ViT-S SMoE-32 | ✓ | | ✓ | ✓ | ViT-S | 241.5M | 10.25M |
| ViT-S SMoE-128 | ✓ | | ✓ | ✓ | ViT-S | 922.3M | 10.25M |
| Virchow | ✓ | ✓ | ✓ | ✓ | ViT-H | 632M | 2B |
| Virchow2 | | | ✓ | | ViT-H | 632M | 1.7B |
| UNI | | | ✓ | | ViT-L | 307M | 100M |

mer occurred either due to increasing the scale of the model from ViT-S to ViT-B or its capacity by switching the feed-forward layer with soft mixture-of-experts at various numbers of experts, whereas variation in the latter was introduced by introducing additional fields of view. The training loss curves plotted along epoch can be found in figure 1. Models from the literature, including Virchow2 (Zimmermann et al., 2024), Virchow (Vorontsov et al., 2024), and UNI (Chen et al., 2024), which utilized ViT-H and ViT-L architectures respectively and were trained on substantially larger datasets, have also been included. These external models were also benchmarked using the procedures described in Appendix B.

# 5 RESULTS AND DISCUSSIONS

## 5.1 MODEL SELECTION

Figures 2 and 3present the complete set of data points of task specific and task agnostic metrics for the ViT-S model trained on the 20× dataset. These sub-figures, including those in Appendix 1, reveal several phases of development in the task-specific and task-agnostic metrics. These figures also help in understanding the intermediate step 3 of algorithm 1, where individual checkpoints from each task-specific and task-agnostic metric pair is extracted from the normalized metric space. Each intermediate checkpoint extracted jointly maximizes benchmark performance and representation quality, and the final model selection maximizes the relative improvement over all tasks, which in this example is the CRC, BACH, and PanNuke 20× tasks.

Looking at this example, the development of model performance in conjunction with the representation rank is observed in the early epochs, followed by a degradation in performance for all segmentation tasks after a certain epoch during training. This suggests that representation ranks are poor indicators of segmentation performance, likely due to the non-linear nature of the task. For the classification tasks, with the exception of BACH, which employs an aggregation function instead of a linear layer (see Appendix B), a clear correlation between the representation rank and performance

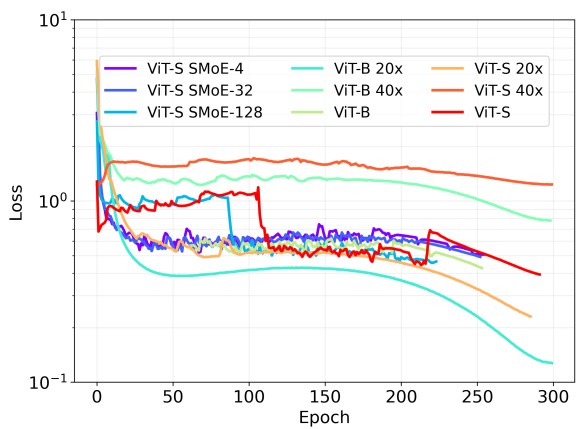

Figure 1: Loss curves for all experiments plotted over epoch.

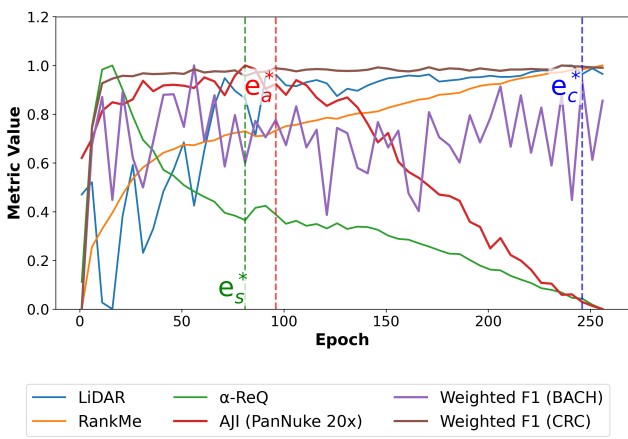

Figure 2: Scaled (between 0 to 1) task-specific and task-agnostic metrics for the case of ViT-S 20×, with the selected models highlighted using vertical lines.

is noticed. Consequently, for the case shown in Figure 3, the classification-best checkpoint occurs much later in the training compared to the segmentation-best checkpoint.

## 5.2 OUT-OF-DISTRIBUTION BENCHMARK PERFORMANCE

Table 2 presents the benchmark performance values for all models described in Table 1, including results for the three different checkpoint types, their corresponding epoch numbers, and the best overall result for each model. Also included are task-specific metric results at the final checkpoint of training for each model.

In figure 1, it is evident that the training loss converges as epochs progress, but in conjunction with the results in table 2, when the task-specific metric results are compared between the final checkpoints and the checkpoints selected using the procedure proposed in this work, the results rarely match, and never exceed those from the selected checkpoints. This shows that training for longer is often detrimental to generalization when it comes to histopathology data, which is in sharp contrast to observations from other data modalities, such as natural language and natural images.

The analysis further reveals that the best-classification model consistently occurs at a later stage of training compared to other checkpoint types, while the best all-round model typically aligns closely with the best-segmentation model. Notably, the models in this study, trained on a single cancer modality with approximately 10 million images, often achieve comparable performance to the provided foundation models, which were trained on pan-cancer data with at least an order of

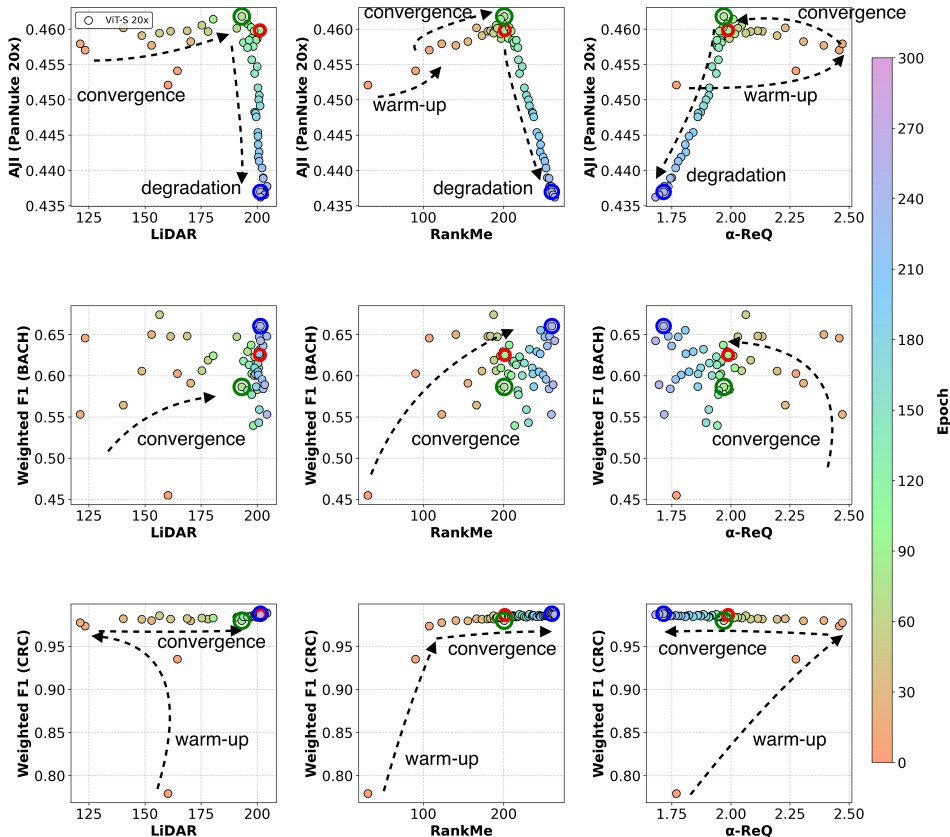

Figure 3: Scatter plots of task-agnostic metrics on the x-axis calculated from the test set against task-specific metrics from out-of-distribution benchmark tasks on the y-axis for the ViT-small model trained on the $20\times$ dataset. Epoch values are used as colours to indicate the evolution of the metric pairs alongside training progression. Dashed arrows show self-interpreted training stages: warm-up, convergence, and degradation. Best models are encircled: all-round ($e_a^*$, red), segmentation ($e_s^*$, green), and classification ($e_c^*$, blue).

magnitude more training samples. For instance, in the MHIST classification task, the best results fall only 3% short of the performance achieved by Zimmermann et al. (2024), despite their model being trained on a substantially larger dataset. In segmentation tasks, the models demonstrate competitive and often superior performance. Specifically, for the PanNuke benchmark (Gamper et al., 2020) at $20\times$ magnification and the MoNuSeg benchmark (Kumar et al., 2019), the best-segmentation and best all-round model frequently outperform the foundation models.

## 5.3 HELD-OUT DOWNSTREAM TASK PERFORMANCE

In the following discussion, only the models trained on multiple FOVs are utilized. Figure 4 presents the AUC performance of three distinct checkpoints for the averaged AUC in the tile-level LUAD subtyping task (Fig. 4a) and the slide-level EGFR classification task (Figs. 4b and 4c). The latter corresponds to aggregation performed over $224 \times 224$ patches ($40\times$ magnification) and $448 \times 448$ patches resized to $224 \times 224$ ($20\times$ magnification), called pseudo $20\times$ in this work. Notably, these tasks were not used in the model selection procedure, ensuring the independence of the individual checkpoint types ($e_a^*$, $e_s^*$, and $e_c^*$) from these tasks.

The LUAD subtyping task results (fig. 4a) indicate that best-segmentation and best all-round model selection criteria can be comparable to or better than best-classification ones in terms of AUC performance, despite the latter typically being trained for longer durations. For the ViT-S model, the bestclassification model selection criteria clearly outperform the other two checkpoints. However, Table 2 reveals that this checkpoint occurs much earlier than the best-segmentation and best all-

Table 2: Model Performance on various out-of-distribution tile-level datasets. Results are rounded to 2 decimal places. **Bold:** best-segmentation model result; **Bold underline**: best-classification model result; **Bold overline**: best all-round model result. †indicates encoders trained on single magnification dataset, including both 20× and 40× encoders. External models have been provided under **Reference**, but have been excluded from the comparative highlighting. Gray highlights cases where the final epoch value result is similar to the best, or exceeds the best task-specific metric among all checkpoints selected for a specific encoder.

| Task-specific metric → | Weighted F1-score | | | Aggregated Jaccard Index | | |
|---|---|---|---|---|---|---|
| Model↓/Benchmark → | MHIST | CRC | BACH | PanNuke | PanNuke | MoNuSeg |
| Magnification → | [5×] | [20×] | [20×] | [20×] | [40×] | [40×] |
| **ViT-S†** | | | | | | |
| $e_a^*$: $96^{20\times}/91^{40\times}$ | - | **0.99** | 0.67 | 0.46 | 0.51 | 0.57 |
| $e_s^*$: $81^{20\times}/91^{40\times}$ | - | 0.98 | 0.63 | 0.46 | 0.51 | 0.57 |
| $e_c^*$: $246^{20\times}$ | - | **0.99** | 0.66 | 0.44 | - | - |
| Final | - | 0.99 | 0.64 | 0.44 | 0.51 | 0.57 |
| **ViT-B†** | | | | | | |
| $e_a^*$: $76^{20\times}/281^{40\times}$ | - | 0.98 | 0.66 | **0.48** | 0.52 | 0.58 |
| $e_s^*$: $76^{20\times}/281^{40\times}$ | - | 0.98 | 0.66 | **0.48** | 0.52 | 0.58 |
| $e_c^*$: $276^{20\times}$ | - | **0.99** | 0.68 | 0.46 | - | - |
| Final | - | 0.99 | 0.61 | 0.47 | 0.52 | 0.57 |
| **ViT-S** | | | | | | |
| $e_a^*$: 166 | 0.84 | 0.98 | **0.68** | 0.46 | **0.53** | 0.57 |
| $e_s^*$: 166 | **0.84** | 0.98 | **0.68** | 0.46 | **0.53** | 0.57 |
| $e_c^*$: 81 | **0.85** | **0.99** | 0.63 | **0.47** | 0.51 | 0.50 |
| Final | 0.80 | 0.98 | 0.67 | 0.44 | 0.52 | 0.56 |
| **ViT-B** | | | | | | |
| $e_a^*$: 51 | **0.85** | 0.98 | 0.65 | **0.48** | 0.51 | 0.57 |
| $e_s^*$: 61 | 0.83 | **0.99** | 0.60 | **0.48** | 0.51 | 0.57 |
| $e_c^*$: 231 | 0.83 | **0.99** | **0.71** | 0.46 | 0.50 | 0.56 |
| Final | 0.82 | 0.99 | 0.66 | 0.46 | 0.50 | 0.57 |
| **ViT-S SMoE-4** | | | | | | |
| $e_a^*$: 111 | 0.84 | 0.98 | 0.64 | 0.47 | **0.53** | 0.59 |
| $e_s^*$: 111 | **0.84** | 0.98 | 0.64 | 0.47 | **0.53** | 0.59 |
| $e_c^*$: 241 | 0.82 | 0.98 | 0.70 | 0.44 | 0.51 | 0.56 |
| Final | 0.83 | 0.98 | 0.64 | 0.44 | 0.52 | 0.56 |
| **ViT-S SMoE-32** | | | | | | |
| $e_a^*$: 131 | 0.84 | 0.98 | 0.67 | 0.46 | **0.53** | **0.60** |
| $e_s^*$: 131 | 0.84 | 0.98 | 0.67 | 0.46 | **0.53** | **0.60** |
| $e_c^*$: 236 | 0.83 | 0.98 | 0.68 | 0.44 | 0.52 | 0.56 |
| Final | 0.81 | 0.98 | 0.60 | 0.44 | 0.52 | 0.56 |
| **ViT-S SMoE-128** | | | | | | |
| $e_a^*$: 166 | 0.84 | **0.99** | 0.65 | 0.46 | **0.53** | 0.59 |
| $e_s^*$: 146 | **0.84** | 0.98 | 0.61 | 0.46 | **0.53** | 0.59 |
| $e_c^*$: 186 | 0.84 | **0.99** | 0.70 | 0.46 | **0.53** | **0.58** |
| Final | 0.81 | 0.98 | 0.56 | 0.45 | 0.53 | 0.57 |
| **Reference** | | | | | | |
| Virchow | - | 1.00 | 0.76 | 0.38 | - | - |
| Virchow2 | 0.88 | 1.00 | 0.80 | 0.48 | 0.57 | 0.58 |
| UNI | - | 1.00 | 0.76 | 0.49 | - | - |

round models, reaffirming that the model performance usually peaks during training. In the slide-level aggregation task (figures 4b and 4c), where classification was performed using ten different train/test splits, we estimate the AUC from the set of predictions that are concatenated from all ten splits. While the AUC performance values do not substantially deviate between checkpoint types, the better performing checkpoint type typically occur in earlier checkpoints rather than later ones, as is seen consistently from the assessments done prior.

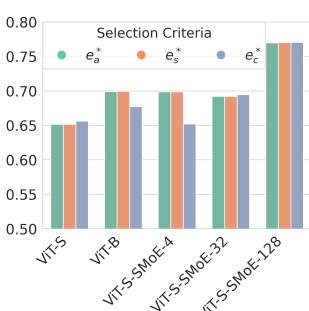 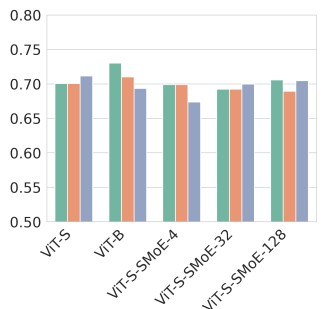 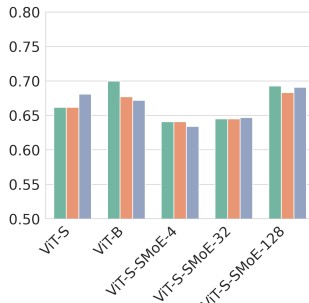

(a) AUC performance for LUAD subtyping at 5× magnification.

(b) AUC performance for EGFR classification at pseudo 20× magnification.

(c) AUC performance for EGFR classification at 40× magnification.

Figure 4: Performance comparison across different magnifications and tasks for encoders trained on patches spanning multiple fields of view.

## 6 CONCLUSIONS

This study addressed the challenge of model selection for histopathology encoders trained in a self-supervised manner. It was shown that training histopathology models for arbitrarily large number of training epochs is actually detrimental to its downstream performance, despite the training loss behavior continuing to monotonically reduce as epochs progress. A model selection procedure for self-supervised encoder training was proposed that combines out-of-distribution task-specific metrics and task-agnostic metrics. In the analyses conducted on several models trained on histopathology data, it was observed that the instance segmentation performance (quantified using the aggregated Jaccard index) was comparable and often exceeded state-of-the-art models from the literature, despite significantly smaller model size, dataset size, and dataset scope in terms of tissue types.

As part of the analyses, model selection criteria were constructed, which differentiated based on the type of benchmark tasks involved in the selection procedure: checkpoints that yield the best results on classification tasks, instance segmentation tasks, and an all-round model considering both task types. These checkpoints were then used to estimate the performance of two held-out tasks measured in terms of AUC: a patch-level LUAD subtype classification task and a slide-level EGFR classification task. While some models showed a preference for one checkpoint type over others, the most performant checkpoints generally occurred mid-way in the training process relative to the final epoch. This is in contrast with the experiences in self-supervised model training on natural images which suggest that longer training leads to better generalization.

A key limitation of this work is that exploring the generalization of each checkpoint is expensive. This is due to the broad range of benchmark tasks, both in terms of scope and field of view, typically encountered in histopathology and the limited ability of current representation quality estimation approaches to estimate instance segmentation performance. Rank estimation approaches may only predict linear probing performance, while typical histopathology classification tasks involve slide-level aggregation of patch-level features, introducing a non-linearity that cannot be directly modeled by an embedding rank. While this limitation was ameliorated in this work by using out-of-distribution benchmark tasks in conjunction with the representation rank, further work is necessary in developing more comprehensive representation quality metrics. This involves a deeper understanding of properties of representations that correlate with the performance of complex downstream tasks beyond classification.

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

# A    DATASETS

## A.1    MODEL TRAINING DATASET

### A.1.1    ENCODER TRAINING DATASET

The dataset consisted of $224 \times 224$ non-overlapping patches extracted from digitized slides of LUAD patients. The scanning was performed at $5\times$, $20\times$, and $40\times$ magnifications, utilizing Leica Aperio AT2 for scanning at $20\times$ magnification (0.5 microns/px), and the Leica Aperio GT450 for scanning at $40\times$ (0.25 microns/px). For each scan at native resolution, a $5\times$ magnification scan is also facilitated by both scanners, i.e., at 2 microns/px. For each magnification, upwards of 5 million samples were selected, extracted from tissue slides stained using hematoxylin and eosin dye. This is collected by randomly sampling 1000 patches per slide at the chosen pixel resolution, performing Otsu thresholding to remove background, i.e., white space.

### A.1.2    TILE LEVEL LUAD SUBTYPE CLASSIFICATION

The LUAD tissue slides used for generating the training set also include slide-level segmentation of commonly found features, specifically regions exhibiting the following characteristics: Acinar, Lepidic, Papillary, Micropapillary, and Solid Tumor. These segmentation labels are utilized to associate each patch with the corresponding majority label, i.e., the class that is predominantly present within the patch. This labeled dataset is then employed for patch-level classification, and the features are most distinctive at lower magnifications or larger fields of view. Consequently, patches extracted at $5\times$ magnification with a resolution of $224 \times 224$ pixels are used for this task. A total of 600,000 images are extracted for this purpose, and their features are obtained for performing classification on top of frozen features. The methodology adopted for this analysis is in accordance with the approach described in appendix A of Radford et al. (2021).

### A.1.3    SLIDE LEVEL EGFR TRAINING DATASET

For the MIL aggregator, a real-world clinical dataset of digitized slides of LUAD patients scanned using an in-house scanning apparatus was chosen, paired with ground truth EGFR mutational status obtained from the IMPACT sequencing panel Cheng et al. (2017). In this case, every tile extracted from the slides is taken after Otsu's thresholding is applied to remove white space. Ten 225/75 train/test splits are utilized from the same 300 set of slides. Only WSIs scanned at 0.25 microns per pixel are considered, as a pseudo-$20\times$ magnification feature set can be obtained from the slides by resizing 448x448 pixel non-overlapping patches back to the encoder's native resolution of $224 \times 224$ pixels.

## A.2    TILE LEVEL OUT-OF-DISTRIBUTION BENCHMARKS

The out-of-distribution generalization of the encoders is evaluated on the following four public histopathology image datasets. For all datasets, minimal preprocessing is applied - resizing/cropping to a uniform $224 \times 224$ input size and converting to RGB format where needed. No stain normalization is applied to preserve the natural variation present in histopathology images.

**PanNuke** Gamper et al. (2020): A large multi-organ nuclei segmentation dataset containing 189,744 nuclei instances across 19 tissue types is used. Images are $224 \times 224$ pixels at $40\times$ magnification. The official 3-fold cross-validation splits are used, with folds 1-2 for training/validation and fold 3 for testing. For benchmarking at $40\times$ magnification, the original image size is retained. At $20\times$, the image is resized to $112 \times 112$ and a $96 \times 96$ center crop is applied. For the Virchow and Virchow2 benchmarking at $20\times$ magnification, a resize of $112 \times 112$ alone is used, and no center crop is performed. This is because the patch size used for both these models is 14.

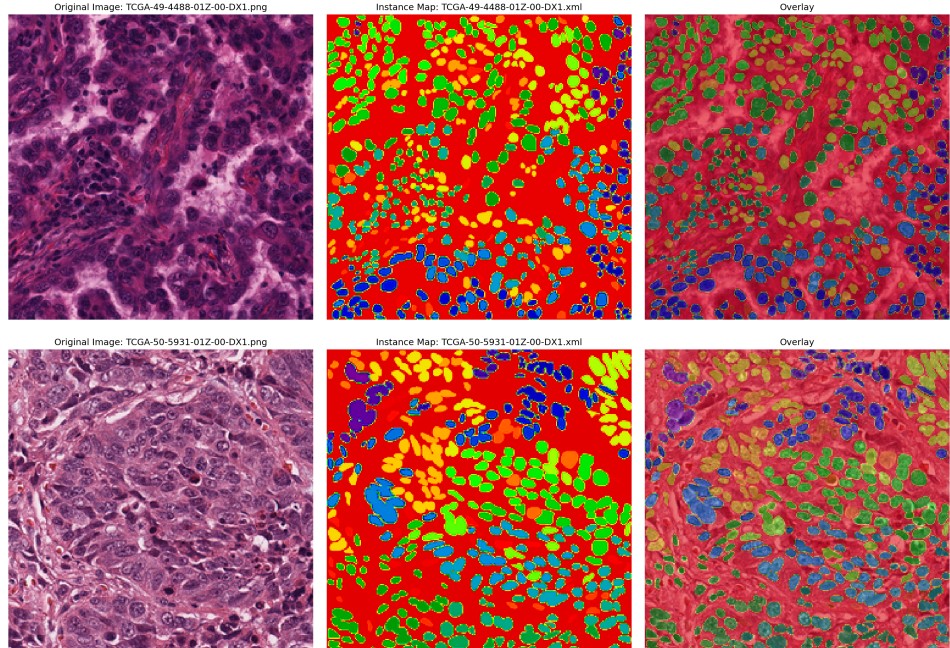

Figure 5: Representative images from the MoNuSeg dataset showcasing the instance level ground truth and overlay with the image.

**MoNuSeg** Kumar et al. (2019): A multi-organ nuclei instance segmentation dataset with 30 $1000 \times 1000$ pixel images at $40\times$ magnification for training and 14 for testing is used. Images span 7 organs including 2 exclusive to the test set. $224 \times 224$ patches are extracted with 75% overlap, maintaining the official train/test split.

**MHIST** Wei et al. (2021): A colorectal polyp classification dataset containing 3,152 images of size $224 \times 224$ pixels, with binary labels of hyperplastic polyp (benign) or sessile serrated adenoma (precursor). We use the official 80/20 train/test split.

**BACH** Aresta et al. (2019): A breast cancer histology image classification dataset with 400 high resolution ($2048 \times 1536$ pixel) images spanning 4 classes: normal, benign, in situ carcinoma, and invasive carcinoma. We use an 80/20 train/test split of the official training data and center crop images to $224 \times 224$.

**CRC** Kather et al. (2018): A nine class Colorectal Cancer dataset containing 100,000 non-overlapping pathes from scanned whole slide images, at 0.5 $\mu$m/px, and at a resolution of $224 \times 224$ pixels. Images are split into an 80/20 train/test split for linear classification. Tissue classes were Adipose, background, debris, lymphocytes, mucus, smooth muscle, normal colon mucosa, cancer-associated stroma, and colorectal adenocarcinoma epithelium.

## B    TRAINING PROCEDURES

### B.1    SELF-SUPERVISED PRE-TRAINING

The DINO self-supervised pre-training methodology Caron et al. (2021) was chosen for training the encoders. The training process utilized a multi-crop strategy with 2 global crops at $224 \times 224$ (scale 0.4-1.0) and 4 local crops at $96 \times 96$ (scale 0.05-0.4), with 8192 prototypes in the projector head. Further data augmentations included random horizontal flipping with a probability of 0.5, random color jittering (brightness, contrast, saturation, and hue) with a probability of 0.8, random grayscale conversion with a probability of 0.01, and Gaussian blurring with a kernel size of 3 and a sigma range of 0.1 to 0.15. Additionally, the second global crop undergoes random solarization with a threshold of 64 and a probability of 0.5. Training was performed at half precision. The

Hyperplastic Polyp →

Sessile Serrated Adenoma →

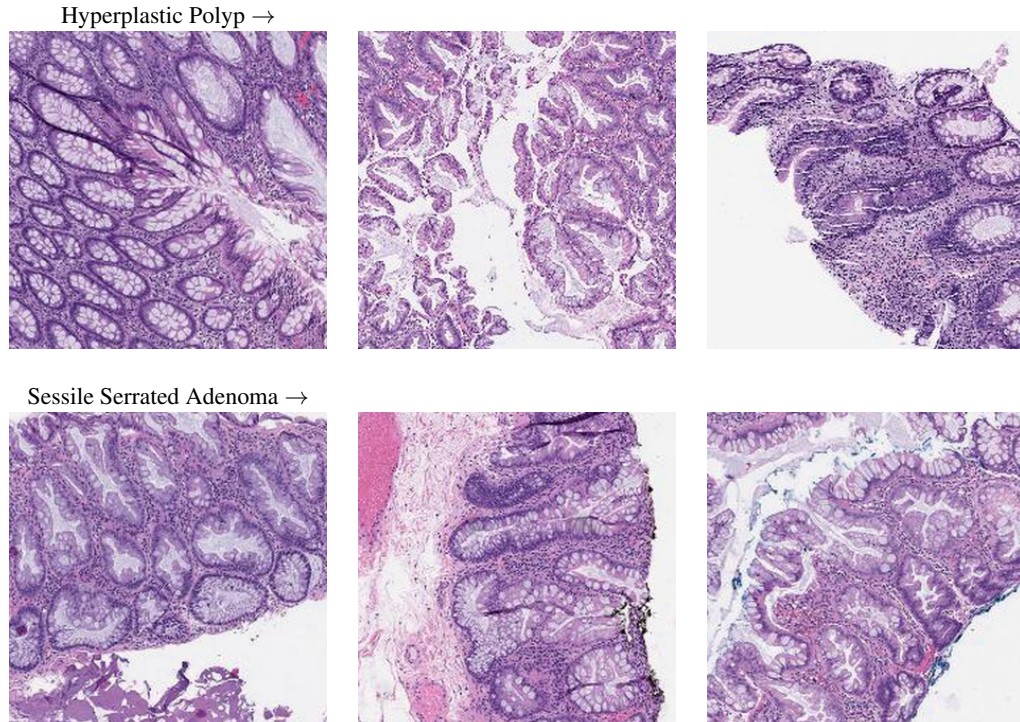

Figure 6: Representative images from the MHIST dataset from the two tissue classes.

AdamW optimizer Loshchilov & Hutter (2019) was employed with a base learning rate of 0.0001, scaled linearly according to batch size, and a minimum learning rate of 1e-06. A cosine learning rate schedule with a 10-epoch warmup period was applied. The momentum parameter of the teacher's weights, updated using exponential moving average (EMA) over the student's weights, started at 0.992 and increased to 1.0 using a cosine schedule. The teacher output's softmax temperature was initialized at 0.01 and reached a final value of 0.04, while the student branch output temperature parameter of the softmax was fixed at 0.1. Weight decay began at 0.04 and increased to 0.4 over 30 epochs. Gradient clipping was set at 0.3, and the last layer was frozen during the first epoch. The center momentum for EMA update was 0.99. Notably, batch normalization was not used in the projector heads, and the last layer of the DINO head was not normalized. Only use the student branch encoder was used for downstream tasks, thus discarding the teacher after training. The loss curves during the training duration are plotted in fig. 1.

## B.2 TILE LEVEL BENCHMARKING

**CRC and MHIST**: The datasets were evaluated using a linear classifier built on top of the pre-trained encoder. This involved a single linear layer, which was trained on the extracted features from the training set. A hyperparameter search was conducted for the weight decay, ranging from 1e-5 to 1e-1. Training utilized the Adam optimizer with a learning rate of 0.001 and employed early stopping with a patience of 5 epochs. The model was trained for a maximum of 50 epochs, using cross-entropy loss and mixed precision training.

**BACH**: For the BACH dataset the classifier incorporated an attention mechanism to handle the larger image sizes ($2048 \times 1536$ pixels). The model processed the image in $224 \times 224$ patches and used attention weights to aggregate features. The training procedure was similar to CRC and MHIST, with the same hyperparameter search and optimization strategy as previous procedures.

**PanNuke and MoNuSeg**: The PanNuke and MoNuSeg benchmarks utilized a modified version of the CellViT model Hörst et al. (2024), focusing on the segmentation task without the classification head. The process involved training the model on the respective datasets for 100 epochs, at full precision using a combined loss function that incorporated cross-entropy, dice loss, mean squared

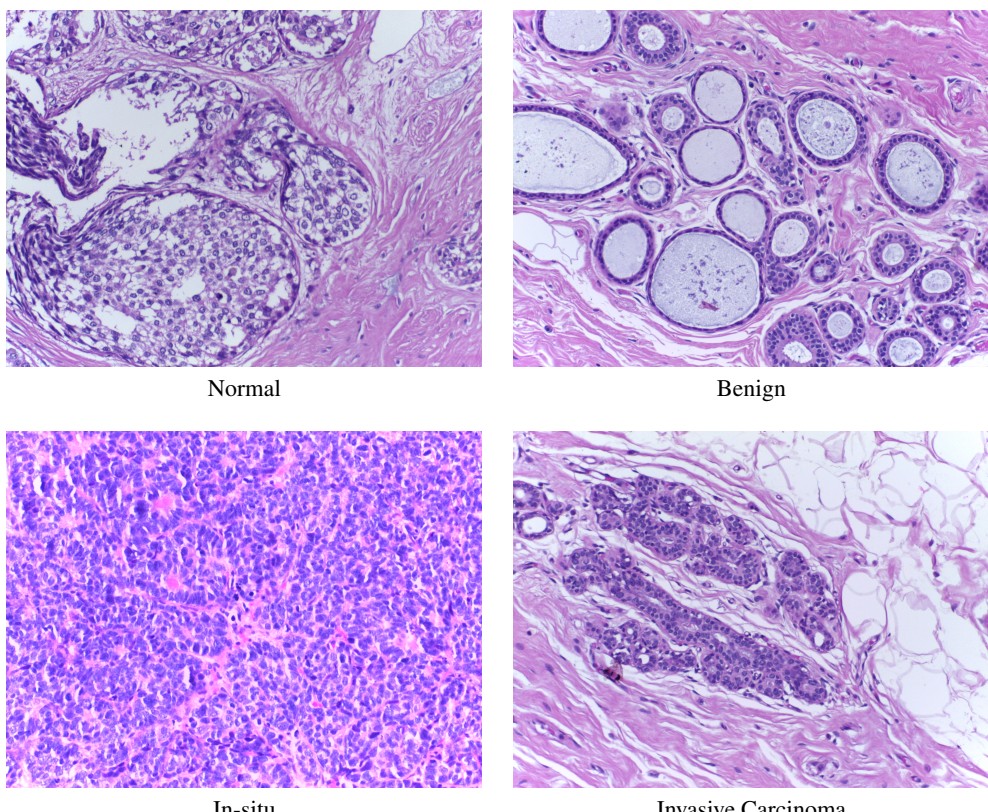

Figure 7: Representative images from the BACH dataset showcasing various tissue classes.

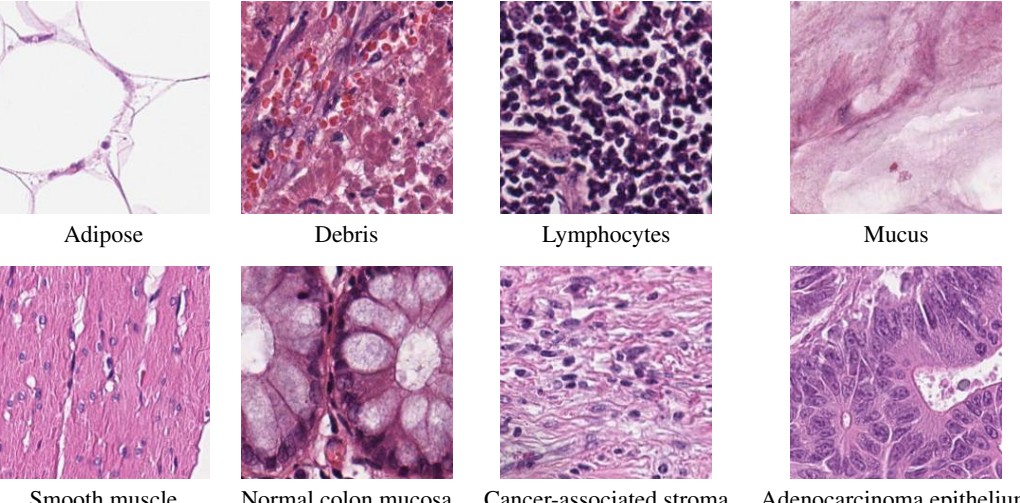

Figure 8: Representative images from the CRC dataset and the various tissue classes excluding the background class.

error, and mean squared gradient error, each weighted based on values found in literature (with weights [1.0, 1.0, 2.5, 8.0] respectively) Graham et al. (2019); Hörst et al. (2024). The model was trained with AdamW optimizer and a custom learning rate scheduler that implemented a warmup phase followed by linear decay. Performance was primarily evaluated using the Aggregated Jaccard Index (AJI) Kumar et al. (2019). For Virchow and Virchow2 benchmarking, the feature maps of the

intermediate ViT blocks were resized from $16 \times 16$ and $8 \times 8$ to $14 \times 14$ and $7 \times 7$ for the $224 \times 224$ and $112 \times 112$ input images. This was done to accommodate the CellViT decoder at the patch size of 14 used by these encoders.

### B.3 DOWNSTREAM AGGREGATOR TRAINING

For the slide-level EGFR prediction task, feature vectors are obtained for all (non-overlapping) patches from the WSI dataset. MIL pooling is then applied on the patch-level feature vector to obtain a slide-level representation of the WSI. This slide-level representation is subsequently used for the binary EGFR classification task. The gated multi-head attention (GMA) Ilse et al. (2018) is used for slide-level aggregation, keeping the encoder frozen during the downstream task.

**Single field-of-view**: Representations come from a single field-of-view of 0.5 microns/pix or 0.25 microns/pix, and $224 \times 224$ pixel patches. They are concatenated as is standard in GMA-MIL to obtain slide-level representations and used for the binary classification task.

**Multiple fields-of-view**: In this case, only WSI scans at 0.25 microns/pix are chosen. Representations come from both 0.5 microns/pix and 0.25 microns/pix at $224 \times 224$ pixel patches. For the former, $448 \times 448$ non-overlapping patches are extracted from the WSI, and the patches are downsampled to $224 \times 224$ to mimic a 0.5 microns/pix patch before being fed to the encoder. For the 0.25 microns/pix, $224 \times 224$ non-overlapping patches are extracted as in the single field-of-view approach. The patches are naively concatenated, and the GMA-MIL aggregator is utilized to obtain slide-level representations which are used for classification.

## C REPRESENTATION QUALITY ESTIMATION

### C.1 ESTIMATION PROCEDURE

To estimate the quality of the learned representations, three metrics are considered: RankMe Garrido et al. (2023), LiDAR Thilak et al. (2023), and $\alpha$-ReQ Agrawal et al. (2022). These metrics are calculated using the test set of the pre-training dataset, which is 20% of the training set size. For LiDAR, 50 augmentations per image are generated to capture the representation behavior under different transformations.

**RankMe:** The RankMe metric is calculated using a random subset of 30,000 embeddings from the encoder to reduce computational complexity. The singular values are obtained through singular value decomposition (SVD) and normalized to sum to one. The final RankMe metric is calculated as the exponential of the entropy of the normalized singular values.

**LiDAR:** A random subset from test features of 1000 samples, each containing 300 augmentations, is chosen, thus giving a total of 300,000 embeddings in a $300 \times 1000$ matrix. In this case, the embeddings are derived from the last but one layer of the Dino projector head which has 256 dimensions. Linear Discriminant Analysis (LDA) is then performed on the reshaped features by estimating the inter-class (between clean images) and intra-class (between augmentations of an image) covariance matrix, and the eigenvalues of the resulting LDA matrix are computed. The eigenvalues are normalized to sum to one, and the LiDAR metric is calculated as the exponential of the entropy of the normalized eigenvalues.

$\alpha$-**ReQ:** The test features are first centered by subtracting the mean, and then the covariance matrix is computed. The eigenvalues of the covariance matrix are obtained and sorted in descending order. The decay coefficient $\alpha$ is then calculated by performing linear regression on the logarithm of the eigenvalues against the logarithm of their indices. The $\alpha$-ReQ metric is calculated as the negative of the slope obtained from the linear regression.

## C.2 RESULTS

### C.2.1 VIT-S-40×

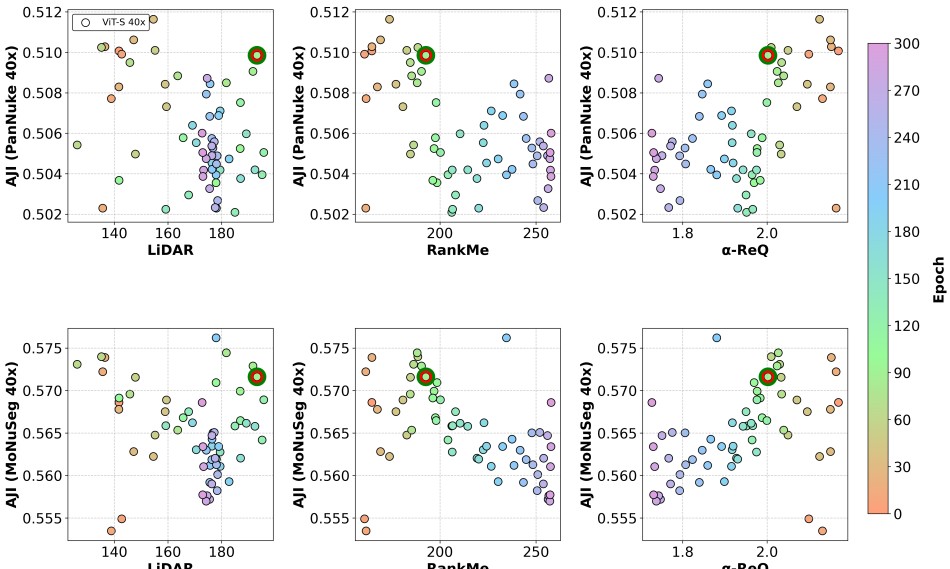

1026
1027
1028
1029
1030
1031
1032
1033
1034
1035
1036
1037
1038
1039
1040
1041
1042
1043
1044
1045
1046
1047
1048
1049
1050
1051
1052
1053
1054
1055
1056
1057
1058
1059
1060
1061
1062
1063
1064
1065
1066
1067
1068
1069
1070
1071
1072
1073
1074
1075
1076
1077
1078
1079

### C.2.2   ViT-S-PMag

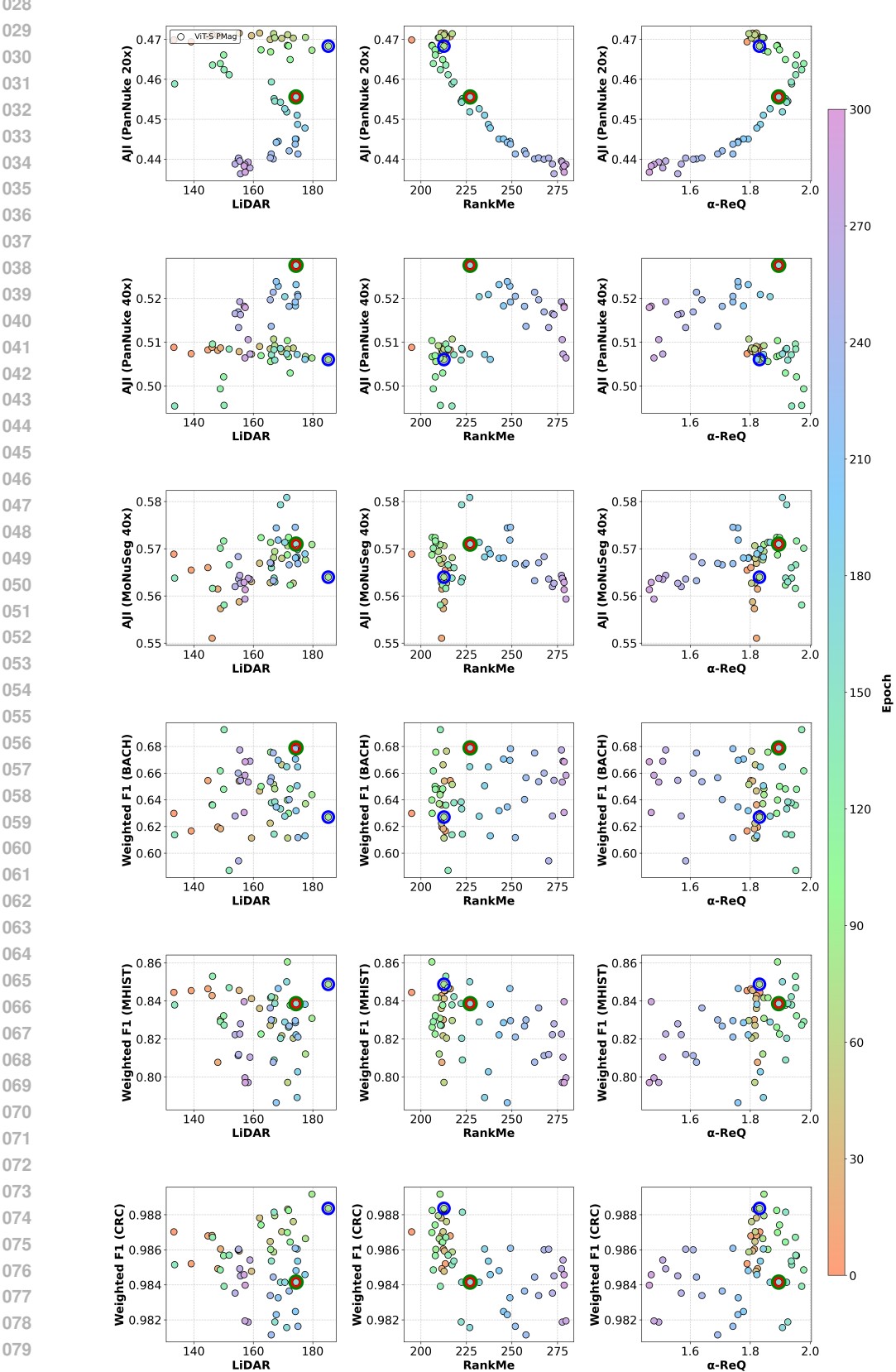

### C.2.3   VɪT-B-20×

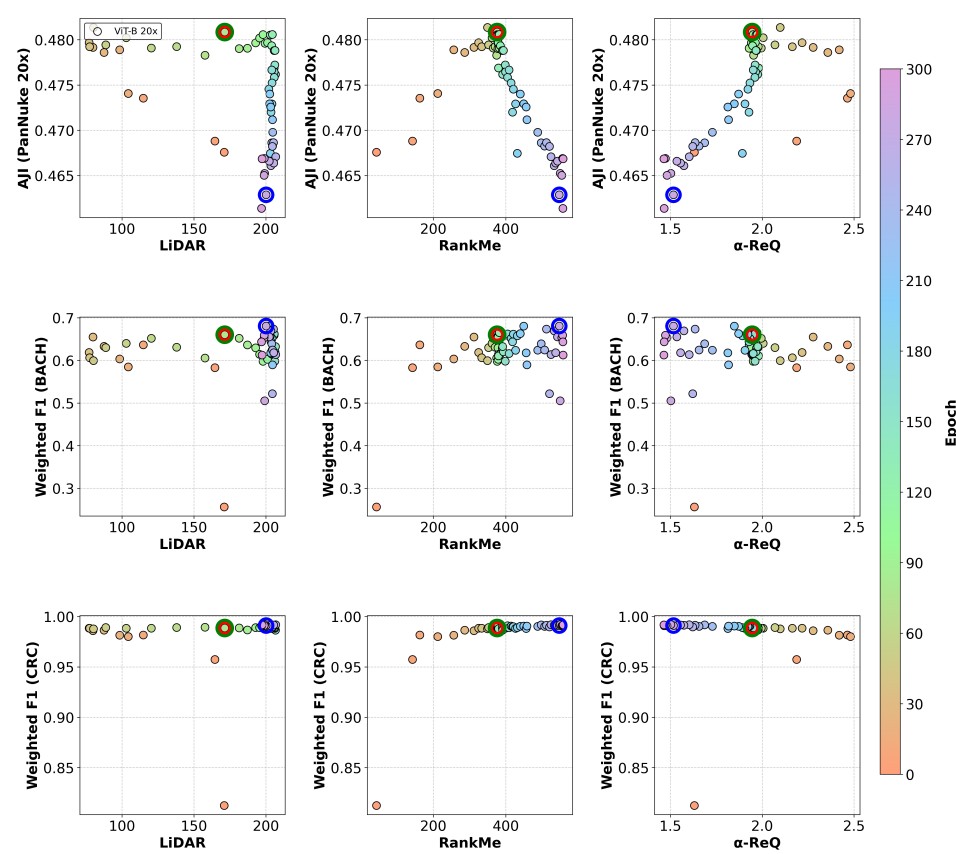

### C.2.4   VɪT-B-40×

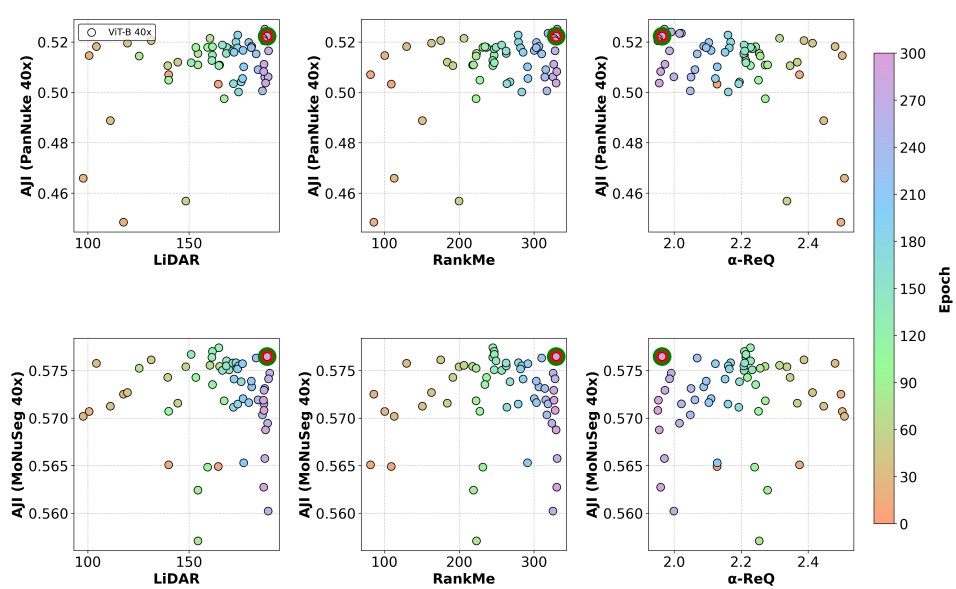

### C.2.5   VIT-B-PMAG

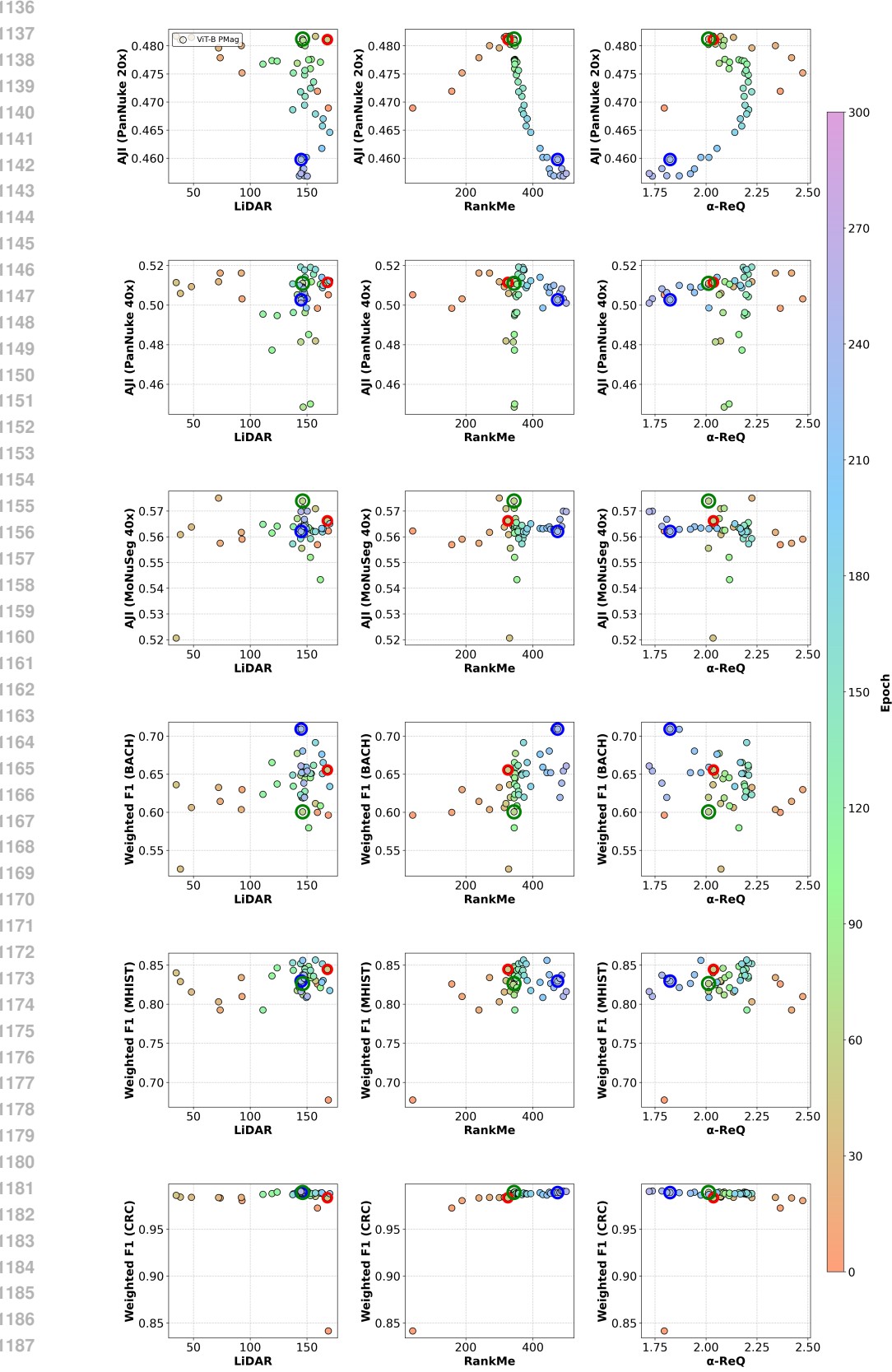

## C.2.6    VIT-S-SMOE-4

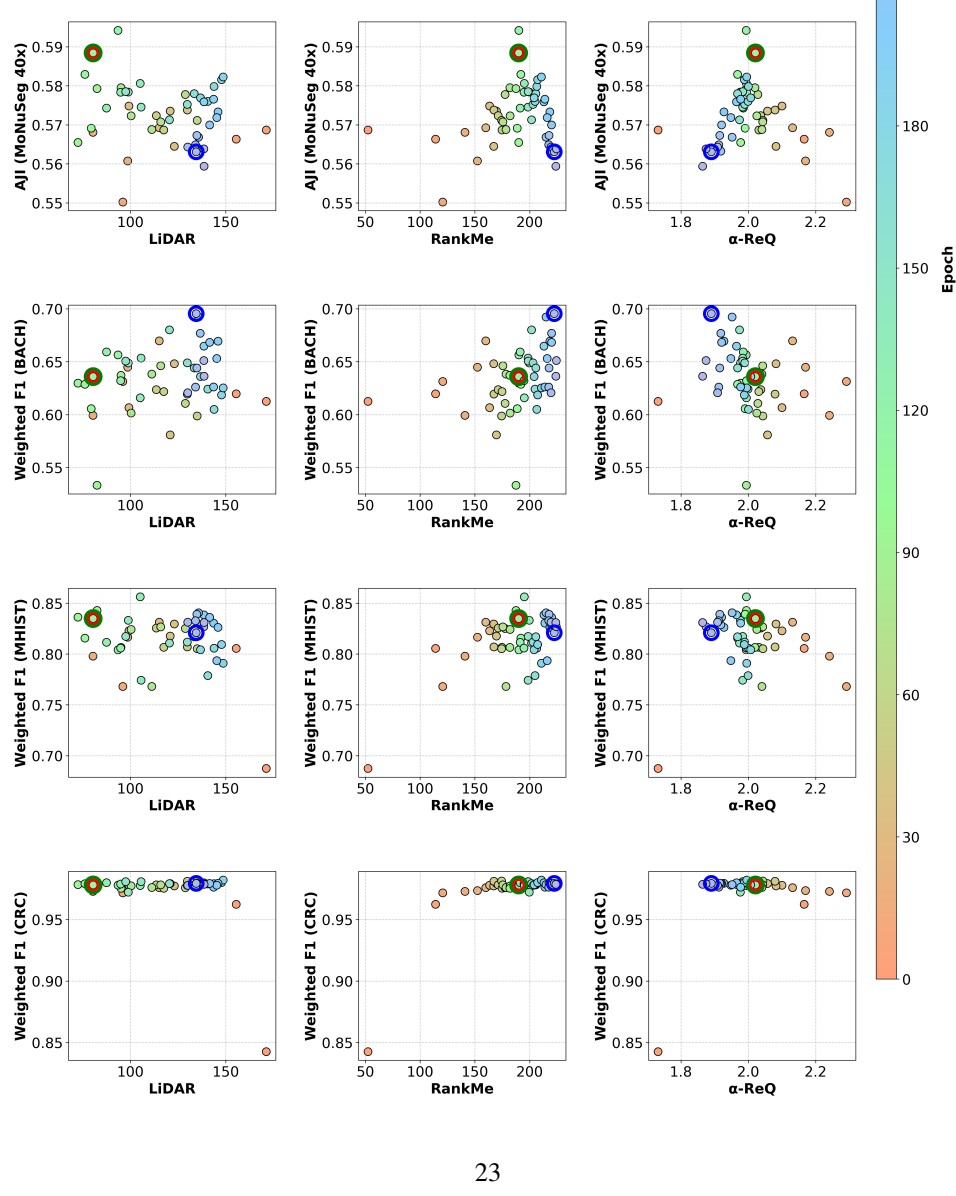

## C.2.7 VIT-S-SMOE-32

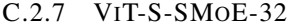

## C.2.8  VIT-S-SMOE-128

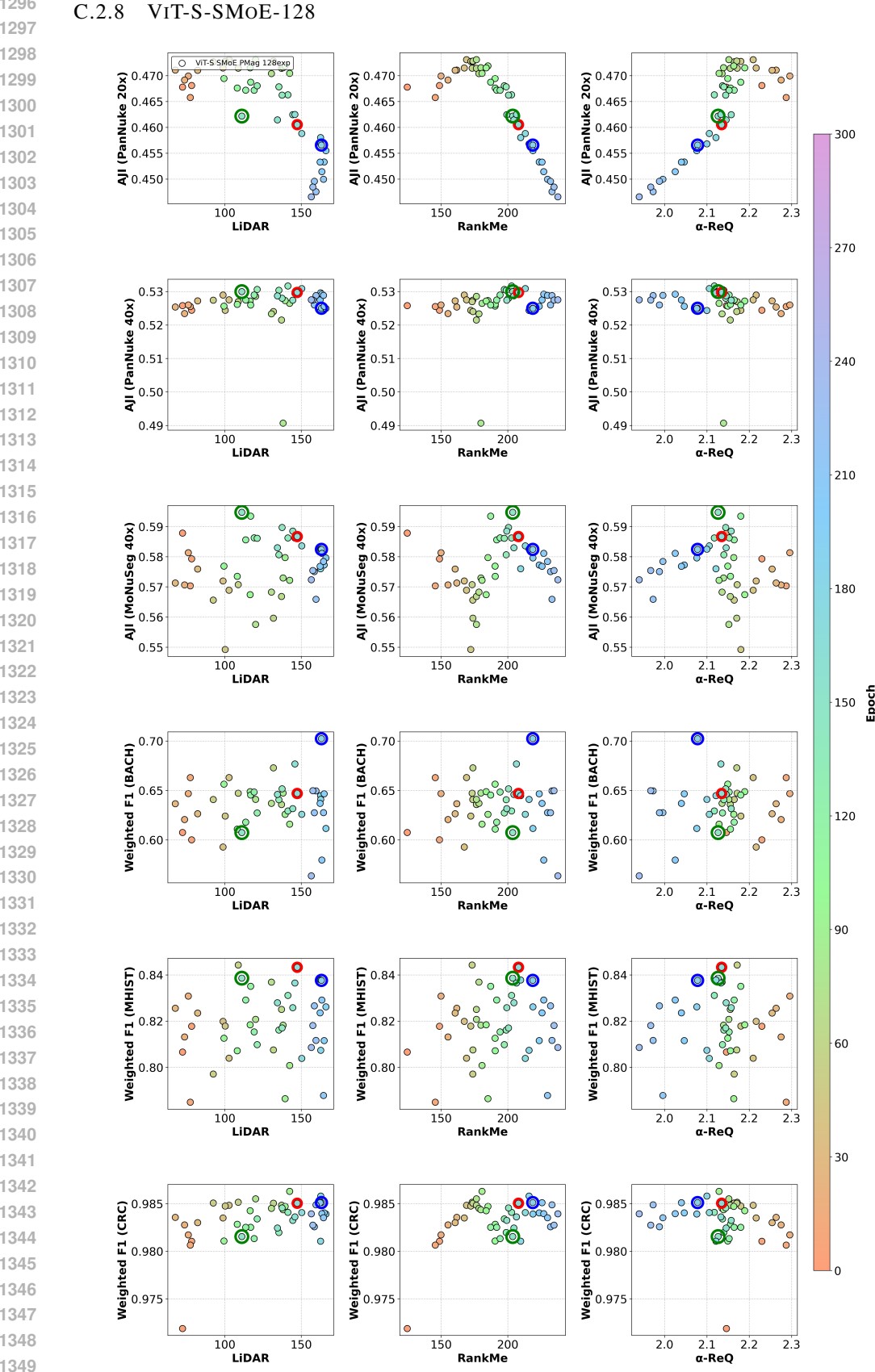

