# OpenReview forum: "A Dual-Metric Approach for Model Selection in self-supervised learning for histopathology"
_ICLR.cc/2025/Conference — Submitted to ICLR 2025_

### Official Review · Reviewer_xPtT · 2024-10-29

**Soundness:** 2
**Presentation:** 3
**Contribution:** 1
**Rating:** 3
**Confidence:** 4

**Summary:**

As outlined in the paper’s contributions, the authors propose a straightforward approach to model selection. This approach involves training various models on different end tasks and evaluating their performance based on classification, segmentation, and task-agnostic metrics. The models are further assessed on an out-of-distribution dataset and a slide-level task to validate their robustness.

More specifically, their model defines a set of $N$ task-specific metrics, $P^{ts}$, and  $M$  task-agnostic metrics, $P^{ta}$. These metrics are evaluated over $E$ epochs, with the $P$ matrices normalized metric-wise using min-max scaling. For each task-specific and task-agnostic metric pair, the epochs that yield the best paired performance are selected. From these selected epochs, the relative improvement for each task-specific metric is calculated, contributing to an overall performance score. Finally, the epoch with the highest relative improvement, $r$, is retained as the optimal model."

**Strengths:**

* Code for reproducibility will be released if the paper is published.
* The related work on task-agnostic quality metrics is well-written and aligns fully with the research scope, as noted by the authors (e.g., "Using these task-agnostic metrics ... is the track followed in this work").
* The proposed algorithm is straightforward and easy to understand.
* The learning process is well represented in Figure 3; however, it is difficult to distinguish $e_a$ (red), $e_s$ (blue), and $e_c$ (green) due to similar color tones.

**Weaknesses:**

* The related work section is overly detailed. While it provides a good overview of self-supervised learning (e.g., SimCLR, CLIP, DINO, iBOT) and multi-scale models (e.g., HIPT, GigaPath), much of the review is not directly relevant to the main focus of the paper. The paper should emphasize metrics rather than the development and history of SSL models. Shortening this section would improve readability.

* For Table 2, three decimal places would improve clarity, as the current values are too similar. Additionally, grouping results by $e_a$, $e_s$, and $e_c$ rather than by models would make the table more informative.

* In Figure 4, a comparison with reference models (e.g., Virchow, UNI) is missing.

While the method is simple overall, it is also somewhat complicated in that there appears to be minimal differentiation among $e_a$, $e_s$, and $e_c$. It would be interesting to show the loss progression on a small validation set to compare with the best epochs. Additionally, there is no clear rule for choosing between $e_a$, $e_s$, or $e_c$.

Other comments:
    A clearer distinction in Figure 2 between task-specific and task-agnostic metrics would be helpful (e.g., using dashed vs. solid lines or separating them). The plot is difficult to read as is.
    (Line 311) Add space between “3” and “present.”
    (Line 368) Ensure consistency with capitalization “Figure”/“figure”.
    (Line 364) Table 2 is cited two pages earlier, which disrupts readability.

**Questions:**

* I am not sure why the algorithm defines relative improvement using only task-specific metrics and not the task-agnostic ones as well. Is it because including the task-agnostic metrics would effectively be equivalent to taking the argmax across all metrics? How much would this change the current estimation of the optimal $e$?

* Here is a list of small details that could be updated:
	* In Algorithm 1, the variable $k$, representing the index of elements in S, is not introduced. It would be helpful to add a note such as $|S| = K$.
	* The expression on line 6 in the algorithm is unclear. It appears that $r$ is a vector; however,  $argmax_n r_n$ returns two indices."

---

> ### Comment · Reviewer_xPtT · 2024-11-26
> **no discussion**
>
> As far as I can tell, the authors have not participated in the rebuttal or discussion.

---

### Official Review · Reviewer_tEBN · 2024-10-31

**Soundness:** 2
**Presentation:** 3
**Contribution:** 2
**Rating:** 3
**Confidence:** 5

**Summary:**

This paper presents a dual-metric model selection strategy for self-supervised learning in histopathology. The main idea is to consider both task-agnostic (like rank) and task-specific metrics (like accuracy) for selecting the best checkpoints. The algorithm can be used to select the best checkpoints for classification, segmentation and all rounds.

**Strengths:**

1. The paper brings an important problem of selecting the best checkpoint in self-supervised training. In most of the existing foundation models for histopathology as well as those for natural images, people still train for a fixed number of epochs and take the last epoch for evaluation.
2. The paper presents detailed analysis of the model's behavior on both downstream tasks and task-agnostic metrics over training epochs, which is valuable.

**Weaknesses:**

1. There are vast arrays of tasks in computational pathology, including ROI classification (by linear classifier or end-to-end finetuning), slide-level classifications, survival analysis, and segmentations. It is not clear enough how does the proposed metrics help on the tasks that's not included in the task metrics. E.g. which metric should user use if they want to select a checkpoint that works the best in slide-level classifications? In Figure 4 b and c, none of the metrics give the best checkpoint that is consistently better than others. Additionally, $e_c^*$ is also not the best for patch-level classification (figure 4a).
2. How does the checkpoint selected by the proposed dual-metric compare to that selected by purely task-specific performance?
3. The training and evaluation are limited. The existing foundation models were typically trained on large data, and evaluated on diverse tasks. While I understand that it's hard to have access to data at that scale, training on LUAD only is limited and it's not clear if the conclusion obtained here is transferrable to larger models trained on larger data size or not. Additiaonlly, the choice of using CRC as one of the evaluation tasks makes the models not discriminative between each other. Rather, TCGA UniformTumor can be better task.

**Questions:**

Given the close performance of between the checkpoints, what are the standard deviations of the performance?

---

> ### Author Response · Authors · 2024-11-25
>
> We thank the reviewer for highlighting the main aim of our work for which the other reviewer do not appear to have understood, and for providing insights on common practices in the field when selecting checkpoints from a self-supervised training run for histopathology (and for natural images). In the following we provide comments to the review:
>
> 1. Weakness 1: We have reworded our conclusion on closer inspection. The earlier conclusion suggested that, by taking a task-type specific checkpoint using our approach, i.e., the $e^*_c$, one could get the best results in classification tasks. This, however, was not the intended conclusion from our work. We were instead trying to motivate that the checkpoint selected using a broad range of surrogate task-types for estimating the task-specific metrics give a more stable model for downstream tasks. This means that the encoder is either as good as, or sometimes better, than the baseline and checkpoints selected using only one task-type, i.e., $e^*_c$. After adding further downstream tasks like TCGA LUAD/LUSC classification at the slide level, among others, we believe that this result will become more clear.
> Therefore, we suggest that a good model selection is possible when a broad range of task-types are used as surrogate tasks for the model selection process. This is seen using from figure 4a and 4c (as pointed out by the reviewer) for the checkpoints selected using classification based surrogate tasks alone, $e^*_c$, which sometimes perform the best, but at times are worse than checkpoints selected using a broad range of task-types, $e^*_a$.
>
> 2. Weakness 2: We request further clarifications on this question by the reviewer. If the reviewer is suggesting we assume only task-specific metrics, setting task-agnostic metrics to zero in our algorithm, then in that case we have not provided this information. This could be a very useful way to showcase the performance of the algorithm, and could serve as a baseline alongside the baseline we will be adding to our results, i.e., results using the final checkpoint of the training run.
>
> 3. Weakness 3: The reviewer here has raised a valuable point, and going forward, we will be training our model on a broad cross-section of tissue types to address this common concern among reviewers on the generalization of our results considering that all our training runs have only being exposed to LUAD data so far.
>
> 4. Question: This critical comment was also raised by reviewer 1, who has requested error bars of the predictions. We will be estimating them and presenting them in the next revision of our work that both covers models trained on multiple tissue types and includes a broader range of publicly available slide-level downstream tasks.
>
>
> (The visibility was wrongly selected in the Readers sections, our apologies)

---

> > ### Comment · Reviewer_tEBN · 2024-12-02
> > **Response to rebuttal**
> >
> > Thanks for authors' reply. In my opnion, we can see the value of this work better if authors can have the weakness and questions addressed. With the current results, the contribution of the proposed method is not clear enough to me. I will keep my score.

---

> ### Comment · Reviewer_tEBN · 2024-11-26
> **no discussion**
>
> As far as I can tell, the authors have not participated in the rebuttal or discussion.

---

### Official Review · Reviewer_Swqz · 2024-11-02

**Soundness:** 2
**Presentation:** 2
**Contribution:** 2
**Rating:** 3
**Confidence:** 4

**Summary:**

The paper propose a model selection approach to find the best checkpoint in self-supervised learning with specific application to digital pathology. The key of the proposed approach is integrating task-agnostic and task-dependent (surrogate task) metric. The authors demonstrate that they can achieve competing performance of larger model trained on more data in certain tasks with way smaller model and data.

**Strengths:**

- Originality: The authors propose an interesting idea to address the issue that the performance boost from self-supervised learning loss does not transfer effectively to downstream tasks. They observed a new phenomenon: different downstream tasks may have different optimal checkpoints that are not necessarily at the optimum of the self-supervised learning loss landscape.

- Quality: The authors trained various models across different model and data sizes to support the proposed approach, providing some evidence for the idea.

- Clarity: The authors used multiple figures to support their observations.

- Significance: The authors present an interesting approach to guide and evaluate the training of self-supervised learning. If more evidence is provided, this approach could be beneficial for training self-supervised learning models.

**Weaknesses:**

- Experiment:
    - Good performance on Table 2 appears to be the direct influence of cherry-pick. As the proposed method involve picking the checkpoint where lead to more performance boost in the surrogate task.
    - Lack of baseline in the Figure 4. If my understanding is correct, the author pretrained model on internal LUAD train, using the surrogate task metric + task agnostic metric to select checkpoint, then testing the performance on internal LUAD test. How does it compare to the directly using self-supervised loss? And how far it is if we directly use foundation model like those shown in Table 2?
    - Lack of generalization baseline: If we apply the same training recipe to pretraining, does the trained model generalize to other cohort like TCGA? In other word, does it do good in TCGA-LUAD/LUSC subtyping? I think the author should at least provide some benchmark on public slide-level dataset
    - What is the subtype statistics in your internal LUAD? Is it balanced/imbalanced? Why did the author report AUC in multi-class classification instead of reporting imbalanced-awared metric like balanced accuracy?

- Methodology: When you only trained on LUAD data, how do you rationalize that data from other organs where the morphology could be very different from LUAD could be a good surrogate task?

- Clarity
  - This paper seems to be prepared in rush, there is a clear typo in line 311 and the first page of the appendix is disorganized
  - The table 2 is confusing due to the under/upperscore, which confuses the current/previous/next rows
  - Methodology: The Step 3 in the proposed algorithm is confusing. Mathematically, How can two matrices with different dimension (i.e., E x N and E x M) be added together then take argmax on one dimension. The authors have to re-write it to better convey the meaning of taking pair of metrics Ts and Ta

- Overall practical use: Since the proposed method is only evaluated on one cancer subtype (as mentioned in the limitation) and need curated surrogate task (which could be irrelevant), I doubt how much gain/ how generalize the proposed approach compared to off-shelf foundation model?

**Questions:**

(1) Can the authors explain the lack of a baselines in the experiment?

(2) Can the authors explain the rationale behind the methodology design? How can tissue with very different morphology help in your pretraining data?

---

> ### Author Response · Authors · 2024-11-25
>
> We thank the reviewer for carefully reading our work and providing insightful comments and observations. We are particularly pleased that the reviewer has pointed out the primary issue with self-supervised learning when applied to histopathology, that the loss objective of self-supervised learning does not correlate to downstream tasks. In the following, we will provide comments and rebuttals to the review:
>
> 1. Experiment Weakness 1: We stress here that the checkpoints selected and presented in table 2 are entirely borne from our proposed model selection approach. For example, it can be observed that none of the selected checkpoints that are based on surrogate tasks purely of one kind (such as classification, or segmentation) are reliably the best performant across all the chosen surrogate tasks. For example, the ViT-B model where the classification optimal checkpoint is selected does not give the highest performance in the MHIST task, this is instead best in the all-round optimal checkpoint, where the surrogate tasks combine both classification and nuclei instance segmentation tasks.
>
> 2. Experiment Weakness 2: It can directly be seen by comparing figures 1 and 2 that while the self-supervised loss monotonically reduces, the performance on the PanNuke segmentation benchmarks at 20x magnification peaks much earlier during training. This observation motivated the work we have presented.
>
> 3. Experiment Weakness 3: We are currently running experiments on TCGA LUAD/LUSC subtyping, including various other tasks.
>
> 4. Experiment Weakness 4: There is imbalance in the data, where Lepidic class is a larger proportion of the train and test data. But this is accounted for in the performance metric computation. The statistics are Acinar (19.2%), Lepidic (31.6%), Papilliary (17.3%), Micropapilliary (16.0%), Solid (15.9%). The AUC reported in our work is the 1-vs-all AUC, which averages binary AUC of one class being positive over all others being treated as negatives, and then averages over all classes considering their individual occurrences.
>
> 5. Methodology Weakness: This is an excellent point. While the morphology of tissues from other organs could be very different, several of the surrogate tasks which have tissues from various organs have reasonable performance metrics for our encoders that have only been trained on LUAD. This suggests that there is a significant overlap in the morphology of LUAD with other tissue types at a patch-level, which is an interesting observation on its own. However, this does not necessarily mean that choosing tissue types from LUAD is sufficient, and therefore, this is a main limitation of our work as mentioned in our scope. In subsequent revisions of our work, we will present encoders trained on a broader range of tissue types to alleviate these concerns.
>
> 6. Clarity Weakness 1: We thank the reviewer for pointing out the typo at line 311, and we request additional guidelines for the first page of the appendix so that we can better match the expectations of the reviewer.
>
> 7. Clarity Weakness 2: We will be completely overhauling table 2 considering the comments from all reviewers to focus on the paper’s main conclusions.
>
> 8. Clarity Weakness 3: We have updated our algorithm to make this step and all the subsequent steps much clearer.
>
> 9. Overall practical use: We will be testing our methodology on encoders trained on multiple cancer types.
>
> 10. Question 1: In the next revision of this manuscript, which will include testing the model on multiple tissue types, will be adding results to reflect a baseline, i.e., selecting the final checkpoint from the training and using it in the downstream tasks presented in figure 4, including public dataset like TCGA LUAD/LUSC tests.
>
> 11. Question 2: The main motivation for our work is described below:
> (1)	 A simple early stopping criteria is not typically utilized when training self-supervised histopathology models.
> (2)	Existing early stopping method that effectively estimate the rank of representation, either via methods like alpha-ReQ, or LiDAR, or RankMe, may predict the performance of linear-classification tasks well, yet, they may not be equally good at prediction tasks commonly encountered in Histopathology, which involves classification using multiple instance learning (MIL), or simultaneous instance segmentation and classification of nuclei in the tissue sample. At the same time, these metrics are useful in that they describe useful properties of representations, such as dimensional collapse, and the relationship between eigenvalues of the representations (for example using the decay rate of the eigenvector of a set of representations).
> These observations led us to combine rank estimation methods, which we characterize as ‘task-agnostic metrics’, and ‘task-specific metrics’ which are performance metrics calculated from a broad range of surrogate tasks. Our approach balances both task-agnostic metrics, and task-specific metrics.

---

### Official Review · Reviewer_Jbv7 · 2024-11-04

**Soundness:** 1
**Presentation:** 2
**Contribution:** 1
**Rating:** 1
**Confidence:** 4

**Summary:**

Model selection is a challenging task in histopathology SSL. The authors proposed a novel model selection approach, and concludes that model selection is important and early stopping may be important for model performance.

**Strengths:**

- Model selection is a very important and relevant task, especially in the era of the explosion of histopathology foundation models.
- Authors included meaningful tasks beyond histological subtyping, including EGFR prediction and nuclei instance segmentation.

**Weaknesses:**

- The proposed model selection procedure appears to be overly simplistic and lacks innovation. It consists of taking the max over many existing task agnostic or task specific existing metrics, but doesn’t include novel metrics to assist model selection or additional insights.
- The benchmarking in the paper is not sufficient. If the author’s claim is that the proposed procedure contributes to model selection, the author ought to compare it to other existing model selection methods. For example, reasonable questions and benchmarks includes: model performance with only use the task agnostic metrics, and model performance on a a task using only a different task specific metric.
- All benchmarks in the manuscript lack error bars.
- The writing of the manuscript can be improved. Some of the notation is not consistent and overcomplicates the proposed method.

**Questions:**

Please address the concerns raised in weakness section, especially on the innovation of the manuscript.

---

> ### Author Response · Authors · 2024-11-25
>
> We thank the reviewer for highlighting the importance of the topic considered by our work, and particularly focus on the diverse range of task types considered by us. In the following, we provide comments and rebuttals:
>
> 1. Weakness 1: We did not intend to propose a new metric because we observed, as can be seen in figure 2 of our work, that task-agnostic metrics like RankMe, or LiDAR, or alpha-ReQ tend to not be useful in predicting downstream task performance of the nature we had chosen. At the same time, these metrics are useful as they measure important properties in the representations, such as rank collapse, or rank diversity. At the same time, we observed that over a range of diverse surrogate tasks such as instance segmentation and linear classification, task-specific metrics like the Aggregated Jaccard Index (AJI) and the F1-score tend to peak at different stages during training. This is why in our work, we proposed an ensemble methodology to combine task-specific and task-agnostic metrics to find a balance between them, which leads to a stable performance in downstream tasks.
>
> 2. Weakness 2: As indicated in our prior response, we show in figure 2 that, when isolated, task-specific and task-agnostic metrics tend to unsuccessfully select a performant model that is suited to downstream tasks. This largely motivated our work to combine any number of task-agnostic metrics that measure representation quality, and task-specific metrics on surrogate tasks.
>
> 3. Weakness 3: We thank the reviewer for pointing this out. We will actively work to introduce error bars of the results in the next iteration of our work, and throughout our benchmarking practices from here on out.
>
> 4. Weakness 4: We appreciate the attention to detail here, but we miss examples of where the notation consistency is violated. By providing some guidance, we can improve the manuscript and make it useful to readers. We request the reviewer to help guide us towards this goal.

---

### Author Response · Authors · 2024-11-25

We thank the reviewers for giving us their valuable time, and providing their valuable comments. These have undoubtedly made several important contributions to the way we want to structure our revision to this manuscript. We cannot presently take all the comments into account to improve our manuscript for this venue, but in this response, we would like to provide some clarifications.

---

### Meta-Review · Area_Chair_gbba · 2024-12-16

**Metareview:**

This paper develops a dual-metric approach for self-supervised learning model selection in histopathology via task-specific metrics and task-agnostic metrics. The proposed method and obtained results show promise in appropriate model selection during histopathological self-supervised learning.

This paper received 1x strong reject and 3x reject from reviewers. The main concerns raised by reviewers regarding this paper centered around limited experimental contribution and novelty. Reviewers agreed that more extensive experiments should be conducted to support the statement of the authors, yet this was not addressed by the authors during the rebuttal.

Given the consensus of reviewers, rejection is recommended.

**Additional Comments On Reviewer Discussion:**

Reviewers mainly raised questions on challenging the novelty of the methodology and experimental contributions. Reviewer Jbv7 commented that the benchmarking in the paper is not sufficient and Reviewers Swqz & tEBN also raised questions regarding the comprehensiveness of experiments the authors conducted. However, these questions are not fully addressed by the authors during the rebuttal.

---

### Decision · Program_Chairs · 2025-01-22

Reject